# Prevalence of Unmet Healthcare Needs in Adolescents and Associated Factors: Data from the Seventh Korea National Health and Nutrition Examination Survey (2016–2018)

**DOI:** 10.3390/ijerph182312781

**Published:** 2021-12-03

**Authors:** Hyeran Park, Jeongok Park

**Affiliations:** 1College of Nursing, Hanyang University, Seoul 04763, Korea; tigress1204@gmail.com; 2College of Nursing, Yonsei University, Seoul 03722, Korea; 3Mo-Im Kim Nursing Research Institute, College of Nursing, Yonsei University, Seoul 03722, Korea

**Keywords:** unmet healthcare needs, adolescent, Korean

## Abstract

Adolescent health is considered the basis of adult health, and the unmet healthcare needs in adolescents are an important issue to be solved. This study, therefore, aimed to explore the prevalence of unmet healthcare needs, and examine its associated factors among adolescents in Korea based on Andersen’s Behavioral Model of Health Services, using data from the Seventh Korea National Health and Nutrition Examination Survey (2016 to 2018). This survey’s data source comprised 12- to 18-year-old adolescents, of which 1425 provided information on their unmet healthcare needs, as well as their predisposing, enabling, and need factors. For statistical analysis, SPSS version 25.0 was used. Descriptive analyses were performed to assess each variable, whereas multiple logistic regression was used to determine the associated factors. The overall prevalence of unmet healthcare needs was 5.5%. The factors that had statistically significant relationships with adolescents’ unmet healthcare needs were: age; stress perceptions; housing types; and perceived health status. Unlike previous studies that presented related factors on vulnerable groups, this study’s results presented unmet healthcare needs and related factors for all Korean adolescents using a national survey dataset. Hence, its findings could provide feedback on current policies, and guide future studies.

## 1. Introduction

Adolescence is considered the healthiest period of the life cycle, with the lowest medical expenses incurred [1]. However, according to the Korea Centers for Disease Control and Prevention (KCDC), adolescent health behavior is a concern because of decreased physical activity, nutritional imbalance, and increased musculoskeletal and spinal abnormalities, which can negatively affect adolescents’ healthy growth and development [2].

Health in childhood and adolescence is considered to be one of the factors that directly or indirectly affects socioeconomic status, such as individual educational achievements, labor market performance, and economic performance in adulthood [3,4]. Therefore, maintaining good health among adolescents is important not only at the individual level, but also at the country level.

The concept of unmet healthcare needs is defined as a state in which individuals do not have access to health care when it is needed [5]. Receiving medical care when needed is one of the important effects on health outcomes, especially in adolescents, since unmet healthcare needs predict poor health conditions in adulthood. In detail, a longitudinal study in the US found that adolescents who reported unmet healthcare needs were more likely to report poor general health, functional impairment, depressive symptoms, and suicidal ideations as adults, compared with adolescents with a similar sociodemographic background, health insurance coverage, and health status, but no unmet healthcare needs [6]. In other words, adolescent health is recognized as the basis of adult health, and unmet healthcare needs in adolescents is an important issue to be solved.

In previous studies, the factors associated with unmet healthcare needs varied across the population. In older adults, economic reasons were the main factors associated with unmet healthcare needs, whereas in the case of adults, the related factors were lack of time and lightness of symptoms [7]. In addition, the prevalence of unmet healthcare needs was high in those who had little use of medical services for disease prevention, such as medical screening [8]. Contrarily, young adults who lived alone or had a weaker sense of belonging to a local community were significantly prone to having unmet healthcare needs [9]. In children and adolescents, family characteristics were found to be the related factors [10,11]. Specifically, support at home from adults, such as parents, were associated with the increased likelihood of adolescents seeking care [11], and higher unmet healthcare needs were reported by children from families with lower household incomes [10]. Moreover, it was reported that health behaviors, such as non-smoking and non-drinking behaviors, were related to healthcare service use [12].

Although adolescents’ health behaviors [13,14] are closely related to those of their parents, as of now, family characteristics, as an aspect of environmental impact, have only been considered in a few studies on adolescent unmet healthcare needs. For an integrated understanding of adolescents’ unmet healthcare needs, there is a need for studies that would consider not only individual characteristics, but also family characteristics, including parents’ health behaviors.

Therefore, this study’s purpose was to explore the prevalence of unmet healthcare needs, and examine its associated factors among adolescents in Korea.

## 2. Materials and Methods

### 2.1. Study Design and Data Source

This study conducted secondary data analysis using the dataset of the Seventh Korea National Health and Nutrition Examination Survey (KNHANES) held from 2016 to 2018. The KNHANES—administered by the KCDC every three years—is a national population-based survey that tracks Koreans’ health. It used a multi-stage stratified cluster sampling of survey regions and households to extract its sample, and collected information on sociodemographic factors, health-related factors, lifestyle factors, and use of medical services [15].

This study was reviewed and approved by the Institutional Review Board of Hanyang University on 4 August 2021 (HYUIRB-202108-002).

Of the 24,269 participants in the 7th KNHANES, 1681 adolescents aged 12 to 18 years were selected for this study. Among them, participants with the following cases were excluded: (1) data with missing values in the main variable (*n* = 135); and (2) answers other than “yes” or “no” to the main question (*n* = 121).

Finally, 1425 adolescents were included in the study. The participants’ selection process is illustrated in Figure 1.

### 2.2. Variables

#### 2.2.1. Dependent Variable

The dependent variable—unmet healthcare needs—was measured by asking the question “Have you ever needed hospital and clinic (excluding dentistry) care (evaluation or treatment) in the past year?” with the answer “yes” being considered as having an unmet healthcare need.

The respondents who indicated they had unmet healthcare needs were asked to provide a further context by answering the question, “Thinking of your experience of unmet needs, why did you not get care?” They could choose from the following eight response options: “Not enough time to visit the hospital,” “Less severe symptoms,” “Financial problems,” “Too far to get to the hospital or inconvenient transportation,” “Not wanting to wait long in the hospital,” “Hard to make an appointment at the hospital,” “Fear of seeing a doctor,” and “etc.” If “etc.” was selected, the respondent was provided with a blank to directly state the reasons.

#### 2.2.2. Independent Variable

Independent variables were categorized into predisposing, enabling, and need variables based on Andersen’s Behavioral Model (Figure 2). This model has been used in several studies as it provides a guide for selecting the factors that may have influenced the decision to use medical services. The model focuses on predisposing, enabling, and need variables that are hypothesized to be related to the use of healthcare services. Predisposing variables include an individual’s demographic and socio-structural characteristics of their environment. Enabling resources comprise the means and abilities of making healthcare services available, along with environmental factors that facilitate individual service use [16]. Need variables are associated with an individual’s level of health, consisting of the needs perceived by individuals, and those evaluated by expert diagnosis [17].

In the current study, predisposing variables included individual characteristics, such as age, gender, stress perception, experiences of smoking and drinking for a year, and influenza vaccination. Respondents’ stress perception tendencies were measured with the question “How much stress do you usually feel in your daily life?” using a 4-point Likert scale: “seriously feel a lot,” “feel a lot,” “feel a little” and “feel very little stress”. The raw data were presented by summarizing these variables into: high—“seriously feel a lot” and “feel a lot”, and low—“feel a little” and “feel very little stress”. With regard to experiences of smoking and drinking for a year, the responses provided were: yes/no. The answer to the question, “Have you been vaccinated against influenza during the past 12 months?” referred to the influenza vaccination status. These variables also included their mothers’ details, such as their education levels (less than middle school or high school/more than college graduation), influenza vaccination status (whether they had been vaccinated during the year), and unmet healthcare needs (whether they had experienced unmet healthcare needs over a year: yes/no), along with familial characteristics: secondhand smoke exposure at home (yes/no).

Enabling variables included: health insurance status (having or not having national health insurance/Medicaid) as individual characteristics; and household income level, residential area (metropolitan city/regional area), and housing type (apartment/detached house/multi-unit house, and others) as familial characteristics. Based on the median income of 50% announced in 2018, the household income level was classified as “high” and “low”, if the amounts were above and below the baseline, respectively.

For the need variables, perceived health status and disease history were used as individual characteristics. The original responses of perceived health status were investigated on a 5-point Likert scale: “Excellent, good, fair, poor, and very poor.” The original responses were re-classified into two categories: good (including excellent, good, and fair) and poor (including poor and very poor). Disease history was adjusted to “yes” when one or more positive reports were made in separate questions of pneumonia, diabetes, allergic rhinitis, atopic dermatitis, asthma, sinusitis, otitis media, urinary tract infection, congenital heart disease, and attention deficit disorder. Their mothers’ perceived health status was used in the need variables as a family characteristic, and adjusted in the same way as the adolescent variable.

### 2.3. Analyses

Statistical analysis was performed using SPSS version 25.0, and the results were considered statistically significant when *p* < 0.05. Descriptive analyses, such as frequency, percentage, mean, and standard deviation, were performed to assess each variable in the study. For univariate analysis, a Chi-squared test, Fisher’s exact test, and *t*-test were performed to identify significant variables associated with unmet healthcare needs. Multicollinearity was confirmed to identify the correlation between independent variables, and the variance inflation factor of all variables was found to be less than 10. Multiple logistic regression analysis was performed to examine the factors associated with adolescents’ unmet healthcare needs. All variables, including the conceptual model, but excluding types of health insurance, were used for the multiple logistic regression analysis. Since the number of adolescents who did not have health insurance was small, it was excluded as a variable from the final multiple logistic regression.

## 3. Results

Table 1 presents the differences in the main variables according to experiences of unmet healthcare needs, based on Andersen’s Behavioral Model. Overall, the prevalence of unmet healthcare needs was 5.5%.

Among predisposing factors, experiences of unmet healthcare needs were significantly different across stress perception (X^2^ = 14.760, *p* < 0.001), smoking experience (X^2^ = 3.900, *p* < 0.001), drinking experience (X^2^ = 5.205, *p* = 0.023), influenza vaccination (X^2^ = 5.137, *p* = 0.023), mothers’ influenza vaccination (X^2^ = 4.818, *p* = 0.028), and secondhand smoke exposure (X^2^ = 6.755, *p* = 0.009). In addition, housing type (X^2^ = 15.787, *p* < 0.001) among enabling factors, and perceived health status (*p* < 0.001) among need factors were significantly associated with experience of unmet healthcare needs.

The results of the multiple logistic regression on unmet healthcare needs are presented in Table 2. Among predisposing factors, age and stress perceptions showed a statistically significant association with unmet healthcare needs. With every 1-year increase in age, the odds of experiencing unmet healthcare needs were multiplied by 0.843 (odds ratio [OR]: 0.843, 95% confidence interval [CI]: 0.722–0.986). Adolescents reporting high levels of stress had a higher odds ratio of experiencing unmet healthcare needs compared to those reporting low levels of stress (OR: 2.054, 95% CI: 1.189–3.546). Among the enabling factors, housing type was significantly associated with unmet healthcare needs. Adolescents living in detached houses had a higher odds ratio of experiencing unmet health care needs compared to those living in apartments (OR: 2.916; 95% CI: 1.592–5.341). Among the need factors, perceived health status was significantly associated with unmet healthcare needs, and adolescents who perceived their health status as poor had a higher odds ratio of experiencing unmet healthcare needs compared to those who perceived it as good (OR: 3.778, 95% CI: 1.734–8.229).

Unmet healthcare needs were reported by 78 adolescents, and their reasons are presented in Table 3. Their main reasons were: did not have enough time to visit the hospital (61.5%); followed by less severe symptoms (32.1%); economic problems (2.6%); and etc., which included unwillingness to wait in the hospital, fear of seeing a doctor, and lack of trust in hospitals (3.9%).

## 4. Discussion

The current study aimed to explore the prevalence of Korean adolescents’ unmet healthcare needs, and examine their associated factors, based on Andersen’s Behavioral Model of Health Services.

An interesting finding in this study was that the prevalence of unmet healthcare needs in Korean adolescents was relatively high compared with those in other countries. The prevalence of unmet healthcare needs in this study was 5.5%, which was similar to that in Canada (5% for boys and 7.2% for girls) in 2014 [19]. Those of the EU [20] and US [21] were 1.8% and 2.9%, respectively. Among countries, the prevalence of unmet healthcare needs seems to be associated with health costs, which are related to the coverage by the national insurance system [22,23]. For example, the US has also reported a reduction in the prevalence of unmet healthcare needs by extending the benefits of Medicaid [24,25].

In general, the reasons for unmet healthcare needs are considered as: availability; accessibility; and acceptability in the literature [26,27,28,29]. Unmet healthcare needs owing to availability issues include long waiting times, unavailability of services when required, and areas with fewer hospitals. In terms of accessibility, unmet healthcare needs arise when individuals are unable to use medical services because of the costs of medical services, or inconvenience in transportation. Whereas availability and accessibility are related to policy reasons, acceptability is about individual perceptions. Therefore, if individuals did not go to the hospital for reasons such as “light symptoms”, “too busy”, “scared to receive medical treatment”, or “don’t know where to go”, it can be linked to acceptability.

In the current study, among the 78 adolescents who experienced unmet healthcare needs, the majority reported that they did not have enough time to visit the hospital, which was an acceptability or availability issue. Among high school students, time pressures and academic stress related to grades, tests, classmates, and career planing were negatively correlated with life satisfaction, and positively correlated with negative emotions [30]. The overheated competition aimed at university entrance exams might have affected adolescents’ voluntary or involuntary abandonment of medical services when appropriate treatment was needed. And in the questionnaire, the answer, “not enough time to visit the hospital” also includes “because the hospital is not open at the time I want.”, which was an availability issue. Visiting the hospital during the daytime could be an academic burden on students because it requires them to opt out of their classes to seek treatment. Problems related to managing time between school schedules and hospital operating hours can force adolescents to voluntarily or involuntarily give up using hospital services. Therefore, expanding evening care in Korea [31] could be an option to increase clinical use, and reduce the prevalence of unmet healthcare needs in adolescents, which can be further improved by building a school-based healthcare system. Strengthening the link between schools and local hospitals has been continually proposed to promote students’ health [32]. Schools arrange regular checkups for students with local hospitals [33]. If possible, a form of extended care, such as regular visiting care or telemedicine performed in schools in conjunction with local hospitals can be considered. By sharing students’ health status between hospitals and schools, students can receive integrated health care from the two institutions. Therefore, if the school and hospital linkage system manages students, students will recognize health as a prominent issue, as well as their academic work. In other words, it can be expected that not only the availability of health care, but also the acceptance, may increase.

Another important finding of this study was that among predisposing factors, stress levels were significantly related to adolescents’ unmet healthcare needs. Adolescents having high levels of stress were more likely to report unmet healthcare needs. Through the literature, it was known that psychological problems were closely related to unmet healthcare needs [7,34,35]. Specifically, adolescents having special healthcare needs coupled with anxiety had increased unmet healthcare needs [34]. A systematic literature review of unmet healthcare needs in Korea also revealed that unmet healthcare needs were high in adults with depression [7]. Similarly, the literature [7,26,36,37,38] has reported that a relevant variable for unmet healthcare needs is perceived health status—a need factor—that is often discussed with negative mental conditions. Negatively perceived health has been interpreted as increasing pain, stress, and depression [39,40], which, in turn, leads to unmet healthcare needs [41]. In this regard, integrated care for both psychological problems and physical health has been proposed as a way to reduce unmet healthcare needs [34,35].

Lastly, housing type, which was a proxy of economic status, was a significant variable among the enabling factors in this study, given that in Korea, people with high incomes live in apartments [42]. According to the Housing Survey by Korea’s Ministry of Land, Infrastructure and Transport, 56.2% of middle-income households and 76.6% of high-income households lived in apartments. Conversely, 50.4% of low-income households lived in detached houses [43], which are widely distributed in rural areas of Korea [44]. The disparity between urban and rural medical facilities is an old issue that has still not been resolved [45]. Therefore, it can be interpreted that adolescents living in detached houses had high unmet healthcare needs owing to low socioeconomic conditions or limited regional characteristics of available hospitals.

In many previous studies, household or individual income levels have been a significant variable in unmet healthcare needs, but in this study, the levels were insignificant. A possible reason could be that families with Medicaid benefits reported higher medical use because of a relatively low burden of medical expenses in Korea [46]. Based on this study’s results, the variable of residence in a detached house may be a proxy for low socioeconomic status (SES) and increased unmet healthcare needs in adolescents. This could mean that there exists a blind spot in Medicaid support, and because of which, it selects targets based only on income levels. Therefore, there is a need to check whether any adolescents having a low economic state are unable to receive medical care, and seriously examine whether choosing Medicaid based only on income levels is the right criteria.

This study has several limitations. The KNHANES was a cross-sectional survey, and thus, it had the limitation of a causal relationship between independent and dependent variables. Therefore, its results need to be interpreted with caution. Since it used a secondary data source, its selection of variables has been limited. In particular, no specific information was available on which healthcare services—such as treatment or examination—were not met. Since the importance and urgency of using medical services could vary depending on treatment and examination, it will be necessary to separate them. In addition, despite the possibility of several events with different reasons for unmet healthcare needs during the course of a year, the participants were asked to choose just one answer. Specific questions to explore each situation will enable a more systematic analysis of unmet healthcare needs. As age increased, the probability of experiencing unmet healthcare needs decreased in this study. However, in previous studies of adolescents, age was not considered a major factor [11]. Therefore, repeated research is needed on the relationship between adolescents’ unmet healthcare needs and age.

Despite these limitations, this study is meaningful in that it explored the overall unmet healthcare needs of adolescents in Korea using a national survey dataset. Many studies have been conducted on unmet healthcare needs for vulnerable groups with special healthcare needs or specific environmental circumstances, but the study of adolescents as a whole can identify what may be missing in previous research and the current policy. Thus, these findings could provide feedback on current policies, and guide future studies.

## 5. Conclusions

This study reported the prevalence of unmet healthcare needs of adolescents in Korea, and analyzed the associated factors using data from the 7th KNHANES. Among the respondents, 5.5% reported insufficient healthcare, with the majority of the reasons related to unmet healthcare needs pertaining to the individual acceptability of medical service use. Major factors associated with Korean adolescents’ unmet healthcare needs include: age; stress perceptions; housing types; and perceived health status.

The findings of this study indicate that measures to increase the acceptability of medical services for Korean adolescents should be prioritized in order to address the unmet healthcare needs of this population in consideration of mental and economic aspects in schools and communities. In addition, as stress perceptions were found to be significantly related to adolescents’ unmet healthcare needs, another opportunity for research is to explore this issue qualitatively.

## Figures and Tables

**Figure 1 ijerph-18-12781-f001:**
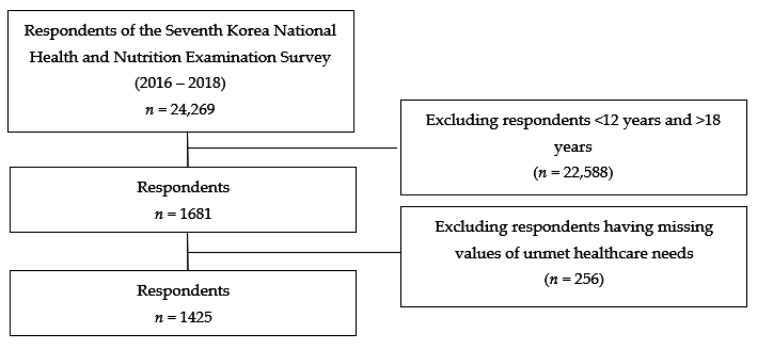
The process of selecting respondents.

**Figure 2 ijerph-18-12781-f002:**
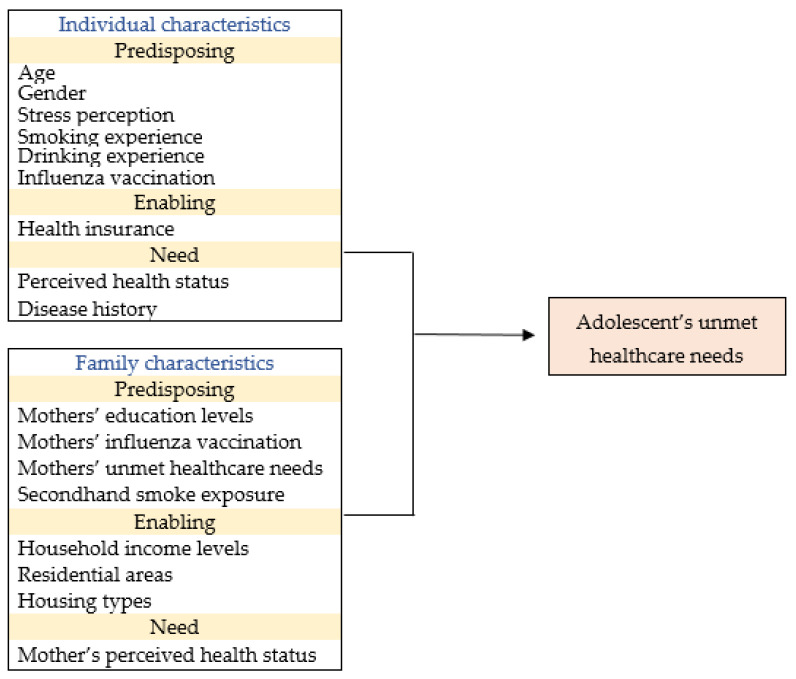
Conceptual framework for the current study, based on Andersen’s Behavioral Model of Health Services. Note: Adapted from “Societal and individual determinants of medical care utilization in the United States” by Andersen, R.; Newman, J.F. 2005, The Milbank Qquarterly, 83, p20 (doi: 10.1111/j.1468-0009.2005.00428.x) [18].

**Table 1 ijerph-18-12781-t001:** Differences of main variables based on Andersen’s Behavioral Model according to experience of unmet healthcare needs (*n* = 1425).

			Adolescents’ Unmet Healthcare Needs					
Variables	*n* (%)		Yes		No		t/χ^2^	*p*-Value
			*n* (%)		*n* (%)			
**Total**	**1425 (100.0)**		**78 (5.5)**		**1347 (94.5)**			
**Individual characteristics**								
**Predisposing factors**								
Age (years): mean (SD)			15.1 (2.1)		14.9 (2.0)		−0.818	0.414
Gender								
Male	738 (51.8)		36 (46.2)		702 (52.1)		1.050	0.306
Female	687 (48.2)		42 (53.8)		645 (47.9)			
Stress perception (*n* = 1424)								
Low	1020 (71.6)		41 (52.6)		979 (72.7)		14.760	<0.001
High	404 (28.4)		37 (47.4)		367 (27.3)			
Smoking experience (*n* = 1424)								
No	1280 (89.9)		65 (83.3)		1215 (90.3)		3.900	0.048
Yes	144 (10.1)		13 (16.7)		131 (9.7)			
Drinking experience (*n* = 1424)								
No	1115 (78.3)		53 (67.9)		1062 (78.9)		5.205	0.023
Yes	309 (21.7)		25 (32.1)		284 (21.1)			
Influenza vaccination (*n* = 1423)								
No	1006 (70.7)		64 (82.1)		942 (70.0)		5.137	0.023
Yes	417 (29.3)		14 (17.9)		403 (30.0)			
**Enabling factors**								
Health insurance								
None	13 (0.9)		2 (2.6)		11 (0.8)			
National health Insurance	1357 (95.2)		73 (93.6)		1284 (95.3)		2.764	0.256 ^b^
National Medicaid	55 (3.9)		3 (3.8)		52 (3.9)			
**Need factors**								
Perceived health status								
Good	1347 (94.5)		65 (83.3)		1282 (95.2)		-	<0.001 ^a^
Poor	78 (5.5)		13 (16.7)		65 (4.8)			
Disease history ^b^ (*n* = 1424)								
No	678 (47.6)		45 (57.7)		633 (47.0)		3.361	0.067
Yes	746 (52.4)		33 (42.3)		713 (53.0)			
**Family characteristics**								
**Predisposing factors**								
Mothers’ education level (*n* = 1230)								
≤Middle school	56 (4.6)		4 (5.8)		52 (4.5)			
high school	528 (42.9)		33 (47.8)		495 (42.6)		1.183	0.554
≥college	646 (52.5)		32 (46.4)		614 (52.9)			
Mothers’ influenza vaccination (*n* = 1230)								
No	956 (77.7)		61 (88.4)		895 (77.1)		4.818	0.028
Yes	274 (22.3)		8 (11.6)		266 (22.9)			
Mothers’ unmet healthcare needs (*n* = 1199)								
No	1116 (93.1)		61 (92.4)		1055 (93.1)		-	0.802 ^a^
Yes	83 (6.9)		5 (7.6)		78 (6.9)			
Secondhand smoke exposure at home (*n* = 1423)	Secondhand smoke exposure at home (*n* = 1423)							
No	1274 (89.5)		63 (80.8)		1211 (90.0)		6.755	0.009
Yes	149 (10.5)		15 (19.2)		134 (10.0)			
**Enabling factors**								
Household income level (*n* = 1423)								
Low	193 (13.6)		15 (19.5)		178 (13.2)		2.432	0.119
High	1230 (86.4)		62 (80.5)		1168 (86.8)			
Residential area								
Metropolitan city	633 (44.4)		36 (46.2)		597 (44.3)		0.100	0.751
Regional area	792 (55.6)		42 (53.8)		750 (55.7)			
Housing type (*n* = 1424)								
Apartment	1027 (72.1)		42 (53.8)		985 (73.2)			
Detached house	252 (17.7)		26 (33.3)		226 (16.8)		15.787	<0.001
Multi-unit house and others	145 (10.2)		10 (12.8)		135 (10.0)			
**Need factors**								
Mothers’ perceived health status (*n* = 1233)								
Good	422 (34.2)		24 (34.8)		398 (34.2)		0.010	0.920
Poor	811 (65.8)		45 (65.2)		766 (65.8)			

^a^ Fisher’s exact test. ^b^ Disease history was adjusted to “yes” when one or more positive reports were made in separate questions of pneumonia, diabetes, allergic rhinitis, atopic dermatitis, asthma, sinusitis, otitis media, urinary tract infection, congenital heart disease, and attention deficit disorder.

**Table 2 ijerph-18-12781-t002:** Results of multiple logistic regression on unmet healthcare needs among adolescents.

	Adolescents’ Unmet Healthcare Needs	
	OR (95% CI)	*p*-Value
Age	0.843 (0.722–0.986)	0.032
Gender (ref = male)		
female	1.184 (0.695–2.016)	0.534
Stress perception (ref = low)		
high	2.054 (1.189–3.546)	0.010
Smoking experience (ref = no)		
yes	1.586 (0.662–3.799)	0.300
Drinking experience for a year (ref = no)		
yes	1.695 (0.820–3.507)	0.154
Influenza vaccination (ref = no)		
yes	0.636 (0.295–1.371)	0.248
Perceived health status (ref = good)		
poor	3.778 (1.734–8.229)	0.001
Disease history ^a^ (ref = no)		
yes	0.599 (0.349–1.027)	0.063
Mothers’ education level (ref = ≤middle school)		
high school	1.244 (0.396–3.907)	0.708
≥college	1.481(0.456–4.808)	0.513
Mother’s influenza vaccination (ref = no)		
yes	0.634(0.263–1.529)	0.311
Mothers’ unmet healthcare needs (ref = no)		
yes	0.864(0.293–2.550)	0.791
Secondhand smoke exposure at home (ref = no)		
yes	1.718(0.831–3.553)	0.144
Household income level (ref = low)		
high	0.785(0.365–1.686)	0.534
Residential area (ref = metropolitan city)		
Regional area	0.997(0.584–1.700)	0.990
Housing type (ref = apartment)		
detached house	2.916(1.592–5.341)	0.001
multi-unit house and others	1.356(0.586–3.137)	0.477
Mothers’ perceived health status (ref = good)		
poor	0.913(0.525–1.588)	0.748

^a^ Disease history was adjusted to “yes” when one or more positive reports were made in separate questions of pneumonia, diabetes, allergic rhinitis, atopic dermatitis, asthma, sinusitis, otitis media, urinary tract infection, congenital heart disease, and attention deficit disorder.

**Table 3 ijerph-18-12781-t003:** Reasons for adolescents’ unmet healthcare needs (*n* = 78).

Reasons for Unmet Healthcare Needs	*n*	%
Not enough time to visit the hospital	48	61.5
Less severe symptoms	25	32.1
Financial problem Etc. ^a^	2 3	2.6 3.9

^a^ unwillingness to wait in the hospital, fear of seeing a doctor, and lack of trust in hospitals.

## Data Availability

Data were obtained from KCDC and are available from https://knhanes.kdca.go.kr/knhanes/sub03/sub03_02_05.do (accessed on 24 June 2020).

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
