# Peer review of "Prevalence of Unmet Healthcare Needs in Adolescents and Associated Factors: Data from the Seventh Korea National Health and Nutrition Examination Survey (2016–2018)"

_ijerph, 2021, doi:10.3390/ijerph182312781_

Round 1

Reviewer 1 Report

Paper presents the interesting study on examination of unmet healthcare needs among adolescents in Korea. I think the article might be suitable for publication. However, I have a number of comments:

  • I would suggest to put the description of the conceptual framework (line 65-74) in the Materials and Methods section and table 3 put in the Results section.
  • The statement: „there were totally 24,269 participants in the seventh survey, whose inclusion criterion comprised 12 to 18-year old adolescent…” (line 89-90) can be confussing, as it seems to be the research sample of the study.
  • The results of the research presented in the Results section are not exactly compatible with data presented in table 2 in terms of OR and CI, for example the odd ratio for Age variable in the text is 84,95% (line 183) and in the table 2 it is 0,843; confidence interval in the text is .77-.99 (line 183) and in the table 2.722-.986.
  • What is the point of presenting the results of multiple logistic regression for variables that turned out to be not statistically significant? Can OR or IC interprated?
  • Not all of the results are discussed in the disscussion section. For example how influenza vaccation correlates with unmet healthcare needs? What is the background of incorporating this variable into the study? Are there any previous research in this context? What are the results?

Author Response

Thank you for reviewing our manuscript. 

We checked  and thought deeply all your comments.

So please see the attachment.

Thank you again.

Reviewer 2 Report

Thank you for submitting your manuscript “Prevalence of Unmet Healthcare Needs in Adolescents and Associated Factors: Data from the Seventh Korea National Health and Nutrition Examination Survey (2016-2018)”. It is an important issue to understand health needs of adolescences as it can be a predictor of adult health care utilisation.

Introduction

What is meant by “smooth” in the statement “Enabling resources comprise the means and abilities of making healthcare services available, along with the environmental factors that make individuals’ use of such services smooth[13]”.

Suggest having a more descriptive label for Figure 1 e.g Conceptual framework for the current study based on the Andersen’s Behavioral Model of Health Services” and add a suitable reference for the model.

Methods

Is there a reference for the KNHANES study to support sampling methodology?

Why were the responses to the question about unmet health needs (when answered yes) predominately about accessing care from hospital? Can adolescences in Korea not access other types of medical services?

Thank you for detailing the variables.

Discussion

Suggest moving Table 3 to be presented in the results, not the discussion. Also, what does “etc” refer to? Please detail explicitly.

The statement “Among countries, the prevalence of unmet healthcare 210 needs seems to be associated with health costs, which are related to the coverage by the national insurance system.” would be stronger if supported with reference(s).

The authors comment on availability, accessibility, and acceptability. These can be barriers to people seeking medical care, but I do not agree that “did not have enough time to visit the hospital” is an acceptability issue. The health issue could just not be that serious. Perhaps a definition of acceptability is needed if it can be subjectively interpreted. I am also unclear on why the questions were focussed on accessing hospital treatment, as mentioned in the feedback above.

Following the comments about predisposing factors stress levels were significantly related to adolescents’ unmet healthcare needs, another opportunity for research would be exploring this issue qualitatively.

Author Response

(The authors gave the same response as above.)
